# Development of Edible Coatings Based on Pineapple Peel (*Ananas Comosus* L.) and Yam Starch (*Dioscorea alata*) for Application in Acerola (*Malpighia emarginata* DC)

**DOI:** 10.3390/foods13182873

**Published:** 2024-09-11

**Authors:** Maria Brígida Fonseca Galvão, Thayza Christina Montenegro Stamford, Flávia Alexsandra Belarmino Rolim de Melo, Gerlane Souza de Lima, Carlos Eduardo Vasconcelos de Oliveira, Ingrid Luana Nicácio de Oliveira, Rita de Cássia de Araújo Bidô, Maria Manuela Estevez Pintado, Maria Elieidy Gomes de Oliveira, Tania Lucia Montenegro Stamford

**Affiliations:** 1Programa de Pós-Graduação em Nutrição, Universidade Federal de Pernambuco, Av. Profª Morais Rego, 1235, Cidade Universitária, Recife 50670-901, Brazil; maria.brigida@ufpe.br (M.B.F.G.); thayza.stamford@ufpe.br (T.C.M.S.); flavia.rolim@ufpe.br (F.A.B.R.d.M.); 2Laboratório de Microbiologia Aplicada, Centro de Ciências da Saúde, Universidade Federal de Pernambuco, Av. Profª Morais Rego, 1235, Cidade Universitária, Recife 50670-901, Brazil; 3Laboratório de Bioquímica, Keizo Asami Institute, Universidade Federal de Pernambuco, Av. Profª Morais Rego, 1235, Cidade Universitária, Recife 50670-901, Brazil; 4Laboratório de Análise de Alimentos, Centro de Ciências da Saúde, UNIESP Centro Universitário, Rod. BR-230, km. 14, João Pessoa 58037-010, Brazil; 5Programa de Pós-Graduação em Ciência e Tecnologia de Alimentos, Universidade Federal da Paraíba, Av. Jardim Universitário, s/n, Universidade Federal da Paraíba—Campus I, João Pessoa 58051-900, Brazil; 6CBQF—Centro de Biotecnologia e Química Fina, Laboratório Associado, Escola Superior de Biotecnologia, Universidade Católica Portuguesa, Rua Diogo Botelho 1327, 4169-005 Porto, Portugal; mpintado@ucp.pt; 7Departamento de Nutrição, Centro de Ciências da Saúde, Universidade Federal da Paraíba, Av. Jardim Universitário, s/n, Universidade Federal da Paraíba—Campus I, Castelo Branco, João Pessoa 58051-900, Brazil; mego@academico.ufpb.br

**Keywords:** agro-industrial waste, edible coating, acerola conservation, quality parameters

## Abstract

Acerola fruit has great nutritional and economic relevance; however, its rapid degradation hinders commercialization. The use of coatings reduces post-harvest biochemical modifications and provides physical and biological protection for vegetables such as acerola. This study developed and characterized an edible coating made from pearl pineapple peel flour (PPPF) and yam starch (YS) to preserve the quality standards of acerola fruits during storage at room temperature and under refrigeration. The edible coating, composed of 4 g of PPPF, 3 g of starch, and 10% glycerol, presented excellent moisture content (11%), light tone (L* 83.68), and opacity (45%), resistance to traction of 27.77 Mpa, elastic modulus of 1.38 Mpa, and elongation percentage of 20%. The total phenolic content of the coating was 278.68 ± 0.45 mg GAE/g and the antioxidant activity by DPPH was 28.85 ± 0.27%. The quality parameters of acerolas were evaluated with three treatments: T1—uncoated fruits; T2—fruits coated with 1% glycerol; and T3—fruits coated with PPPF-YS. The T3 treatment reduced the weight loss of stored acerolas, maintaining the light and bright color of the fruits, and delayed the decrease in soluble solids, especially in refrigerated fruits. Therefore, edible coatings based on pineapple flour and yam starch are effective technologies for controlling the physical and physicochemical parameters of acerolas during storage, benefiting the post-harvest quality of this fruit.

## 1. Introduction

Acerola (*Malpighia emarginata* DC.) is a tropical fruit that thrives in tropical and subtropical climates, known for its rich composition of phenolic and bioactive compounds, such as anthocyanins, carotenoids, flavonoids, and vitamin C [1]. Brazil plays a significant role in the international market, exporting acerola derivatives in fresh, frozen, and pulp forms [2,3]. However, as a climacteric fruit, acerola exhibits high respiration rates and has an extremely short shelf life after harvest, limiting the available time for transportation and storage and resulting in substantial post-harvest losses [4,5]. These characteristics highlight the need for biotechnological strategies to extend acerola’s shelf life. Among the various biotechnological approaches, the application of edible coatings has emerged as a promising alternative for preserving tropical fruits, including acerola [6,7].

Another fruit with global prominence and high productivity is the pearl pineapple (*Ananas comosus* L.). Widely used in the food industry, pearl pineapple generates significant waste (peels, stalks, crowns, and cores) that can become potential sources of contamination if not properly disposed of. Pearl pineapple peel is a good source of fiber and contains other nutrients such as lipids, proteins, phosphorus, vitamin C, and carotenoids. This waste can be used in the formulation of edible coatings, providing both nutritional and functional properties [8]. Starch, one of the most widely used biopolymers in developing fruit coatings, stands out due to its physicochemical properties, such as water absorption capacity and solubility, and its rheological properties, including gel formation. These qualities make starch a key component in biodegradable coating bases, often used in combination with lipids, proteins, or other polysaccharides [9,10].

While cassava, corn, and potatoes are the most common sources of starch globally, using non-conventional sources offers benefits such as waste reduction, promotion of sustainability, and encouragement of regional technological development. Yam (*Dioscorea alata* L.) is an example of a non-conventional starch source, particularly important in Brazil’s Northeast region, but still underexplored in terms of its properties and applications [11,12]. Edible coatings made from agro-industrial waste help reduce the environmental impact of improper disposal while showing promising results in fruit preservation, contributing to sustainability in agribusiness [13,14]. These coatings are typically derived from renewable sources, making them eco-friendly and cost-effective. They also serve as alternative packaging materials that help mitigate the environmental impact of conventional packaging [15].

In this context, this study aimed to develop and characterize an edible coating produced from pearl pineapple peel flour (*Ananas comosus* L.) and yam starch (*Dioscorea alata* L.), with the objective of preserving the physicochemical and microbiological quality of acerola (*Malpighia emarginata* DC.) during storage at both room and refrigerated temperatures. It is believed that combining pearl pineapple peel flour with yam starch enables the development of a sustainable, low-cost edible coating capable of extending acerola’s shelf life. Additionally, the fiber and nutrients found in pearl pineapple peel, as documented in the literature, may contribute functional claims that benefit consumer health and promote sustainable and technological actions.

## 2. Materials and Methods

### 2.1. Materials

Acerola (*Malpighia emarginata* DC.), pineapple (*Ananas comosus* L.), and yam (*Dioscorea alata* L.) were purchased from a local sustainable ecological farming fair certified by the City of João Pessoa (PB, Brazil). Acerolas with 100% red skin, pearl pineapples with 30% yellow skin, and yams free of visible damage were selected based on visual analysis.

### 2.2. Methods

#### 2.2.1. Yam Starch Extraction (YS)

Yam tubers were peeled, cut into cubes, and immersed in sodium metabisulfite (7 g/L) for 24 h. To extract the starch, the yam was blended (MONDIAL Turbo Power L-99-F, Conceição do Jacuípe, Brazil), filtered through voile fabric, and allowed to decant. After 24 h, the supernatant was discarded, and the starchy material was dried on a stainless-steel tray in an oven (Tecnal TE-371, Kyoto, Japan) at 60 °C for 24 h. Finally, the dried material was sieved through a 100-mesh screen to obtain starch powder.

#### 2.2.2. Preparation of Pearl Pineapple Peel Flour (PPPF)

The pineapple peels were separated from the pulp with the aid of a knife (chef), cleaned, sanitized, and then immersed in a sodium metabisulfite solution (7 g/L) for 5 min. The peels were crushed in a mini-processor (Black&Decker, model MP120, Uberaba, Brazil), subjected to prior freezing (−40 ± 1 °C), and dried in a freeze-dryer (LIOTOP, model L101, São Paulo, Brazil) for a period of 24 h (54 uHg; −57 °C; 226 Vac). The material was then crushed in a mini processor (Black Decker) for 10 min and subsequently sieved through a 100-mesh screen to make the flour uniform.

#### 2.2.3. Characterization of PPPF

##### Physical Analysis and Technological Properties of PPPF

Color was measured in triplicate using a digital colorimeter (Konica Minolta CR 400) based on the CIELAB system, assessing chromatic coordinates L* (luminosity from black (0) to white (100)), a* (green (−a) to red (+a)), and b* (blue (−b) to yellow (+b)).

Granulometry was determined by passing 100 g of PPPF through sieves (16, 18, 20, and 35 mesh) in a mechanical agitator (Produtest T, São Paulo, Brazil) with 15 min of vibration at maximum intensity. The fractions collected from the sieves were weighed, and the results were expressed as percentages according to Equation (1).
(1)%Retention:Pf−Pi/Pa×100
where Pf = mass (g) of the sieve with the retained content; Pi = mass (g) of the sieve; and PA = mass (g) of the sample.

Water Absorption Index (WAI)

A total of 1 g of PPPF was mixed with 10 mL of distilled water and stirred for 30 s in a vortex (KASVI, model K45-2810, São José dos Pinhais, Brazil). The mixture was left at room temperature (25 ± 0.5 °C) for 30 min, then centrifuged at 2000× *g* for 30 min. The supernatant volume was measured, and the Water Absorption Index (WAI) was calculated using the initial water volume and the supernatant volume, as described in Equation (2).
(2)WAI=Water absorbed by sample (g) / Weight of sample (g)

Water Solubility Index (WSI)

To analyze flour water solubility, 600 mg of PPPF was mixed with 30 mL of water and stirred for 30 min at 55–85 °C. The mixture was then centrifuged at 2058× *g* for 15 min. The supernatant was dried at 130 °C for 12 h, weighed, and the solubility was calculated as a percentage of the mass of the dry supernatant relative to the initial flour mass, as per Equation (3).
(3)$$WSI=\frac{Evaporation\;residue\left(g\right)}{Sample\;weight\;\left(g\right)}\;\times 100$$

Swelling volume (SV)

The swelling volume (SV) was determined by adding 30 mL of distilled water to 1 g of PPPF in a 100 mL graduated cylinder. After stirring for 30 min with a magnetic bar, the sample was left to stand for 15 h for complete decantation. The SV was expressed in mL/g of dry matter by observing the volume occupied by the sample in the graduated cylinder [16]. 

##### Physical–Chemical Analysis of PPPF

Water activity at 25 °C was measured with a calibrated water activity meter (Decagon, Pawkit model, Washington, DC, USA). The acidity, pH, moisture, ash, protein, and total fiber content of pearl pineapple peel flour was analyzed. Lipid content was assessed [17], and total carbohydrate content was calculated by subtracting the sum of moisture, ash, lipid, protein, and crude fiber from 100.

##### Determination of Total Phenolic Compound Content

Total phenolic compounds were analyzed with minor modifications [18]. A methanolic extract was prepared [19], and 20 µL of this extract was mixed with 100 µL of Folin–Ciocalteu reagent (10%, *v*/*v*) and 80 µL of sodium carbonate (75 g/L) in a 96-well plate. After 90 min of incubation in the dark at 25 ± 0.5 °C, absorbance was measured at 765 nm using a spectrophotometer (BioTek μQuant Biospectro, Winooski, VT, USA). Total phenols were quantified using a gallic acid standard curve and expressed as milligrams of gallic acid equivalent per gram of sample (mg GAE/g).

##### Determination of Antioxidant Activity

To assess the antioxidant activity of PPPF, the % elimination of ABTS^•+^ (2,2-azino-bis-3-ethylbenzthiazoline-6-sulfonic acid) and DPPH (2,2-diphenyl-1-picrylhydrazyl) radicals was analyzed. For the ABTS^•+^ radical scavenging assay [20], a methanolic extract was prepared [19]. An ABTS solution was made by dissolving 7 mM ABTS in 2.45 mM potassium persulfate (K_2_S_2_O_8_) and incubating it in the dark at 25 ± 0.5 °C for 16 h. This solution was diluted in ethanol (A.R. grade) to achieve an absorbance of 0.70 ± 0.02 at 734 nm. Then, 20 µL of the liquid extracts (10%) was mixed with 1 mL of the ABTS solution and incubated for 6 min. Absorbance was measured at 734 nm using a spectrophotometer (BioTek μQuant Biospectro, Winooski, VT, USA). ABTS^•+^ radical elimination was calculated as a percentage based on radical inhibition, according to Equation (4).
(4)ABTS radical inhibition(%)=[1−(Aa−Ab/Ac)×100
where *Aa* = absorbance of the sample + ABTS solution; *Ab* = absorbance of the sample without the ABTS solution; and *Ac* = absorbance of the blank control without the sample.

For the DPPH radical scavenging assay [21], 1 mg of the extract was dissolved in 1 mL of methanol (A.R. grade), then serially diluted to concentrations ranging from 15.625 µg/mL to 1000 µg/mL. In a 96-well plate, 40 µL of each dilution was mixed with 250 µL of DPPH reagent. After 30 min of incubation in the dark at 25 ± 0.5 °C, absorbance was measured at 517 nm using a spectrophotometer. DPPH radical scavenging activity was calculated using Equation (5).
(5)DPPH (%)=(Ab−Aa / Ab) × 100
where *Aa* = absorbance of the sample + DPPH solution, and *Ab* = absorbance of the blank control without the sample.

#### 2.2.4. Preparation of Edible Coating

The development of the coating consisted of mixing PPPF with distilled water (4 g/100 mL, *w*/*v*), followed by the addition of YS (3 g), heating under constant stirring at 80 °C for 15 min, the addition of glycerol (10%, *w*/*w* in relation to PPPF), heating under stirring at 80 °C for 15 min, and vacuum filtration through Cicatrisan gauze^®^ (Americana, Brazil).

#### 2.2.5. Characterization of Edible Coating

##### Thickness

The coating thickness was measured with a digital micrometer (Multicomp Pro, Shanghai, China) at five random points. The final thickness was calculated as the average of these measurements.

##### Moisture

A 0.5 g sample of the coating was dried in an oven at 105 °C for 24 h, which is sufficient time to achieve a constant mass. The final mass was then recorded, and the percentage of coating was calculated [22]. 

##### Color and Opacity

The color of the edible coatings was measured with a Konica Minolta CR 400 colorimeter, following CIELAB parameters (L*, a*, b*). Opacity was determined by measuring the coatings on both white and black backgrounds and calculated using Equation (6).
(6)OP=(OPB / OPW)  × 100
where *OPB* = opacity of the coating against a black background and *OPW* = opacity of the coating against a white background. The standard values used on a white background were 84.67, 0.55, and 0.68 for L, a, and b, respectively. 

##### Water Solubility

A sample approximately 2 cm in diameter was weighed to determine the initial weight. Simultaneously, the moisture content of the coatings was measured to calculate the initial dry mass (im). The samples were then immersed in 50 mL of distilled water in an Erlenmeyer flask and placed in a refrigerated shaker incubator (Tecnal TE-424, Piracicaba, São Paulo) at 25 ± 0.5 °C for 24 h. Afterward, the solution in each flask was discarded, and the solid coating was separated. The final dry mass of the non-solubilized sample (fm) was obtained by drying at 105 °C until a constant weight. Water solubility was calculated using Equation (7).
(7)% MS=mi−mf ÷ mi×100 
where %*MS* = percentage of solubilized material; *mi* = initial dry mass of the sample; and *mf* = final dry mass of the non-solubilized sample.

##### Mechanical Properties

The mechanical properties were determined with modifications based on [23]. Tensile strength (MPa), elastic modulus, and elongation percentage (%) were measured using a texture analyzer (Brookfield CT3 Texture Analyzer, Middleboro, MA, USA). The gripper speed was set at 5 mm/min, with an initial distance of 40 mm between the grippers. The test specimens measured 2.5 cm in width and 7.5 cm in height.

##### Determination of Total Phenolic Compound Content and Antioxidant Activity

The total phenolic content was evaluated as outlined in Section Determination of Total Phenolic Compound Content, and antioxidant activity was assessed according to Section Determination of Antioxidant Activity.

#### 2.2.6. Evaluation of Quality Parameters of Coated Acerola Fruits during Storage at Room and Refrigerated Temperatures

The fruit quality parameters were evaluated under three treatments: T1—negative control (uncoated fruit), T2—positive control (fruit coated with 1% glycerol), and T3—fruit coated with the test coating (PPPF-YS). Semi-ripe acerolas at stage 2 (75% of the surface in orange to red color) were selected, ensuring no visible microbial infections or physical damage. The fruits were sanitized in sodium hypochlorite solution (20 mL/L) for 10 min, rinsed with running water, and air-dried at room temperature (25 ± 0.5 °C) for 1 h. After drying, the fruits were immersed in the respective coating treatments (T1–T3) for 3 min, then naturally dried on voile fabric-lined trays. They were distributed into Petri dishes (150 × 25 mm) labeled with their treatment and stored at room temperature (25 ± 0.5 °C) for 4 days, or under refrigeration (5 ± 0.5 °C) for 8 days. The experiment was conducted in triplicate, with each tray containing 60 g of acerola for each treatment, temperature, and storage time. Quality assessments were conducted at time 0 (immediately after coating) and at 2-day intervals for each storage temperature, based on microbiological, physical, and physicochemical characteristics.

##### Microbiological Analysis of Coated Acerolas

During storage at different temperatures, the fruits were analyzed for *E. coli* and *Salmonella* spp. [24] using the 3M™ Petrifilm™ rapid method (3M do Brasil Sumaré/SP).

For each group (T1, T2, T3), at each storage temperature and time point (initial and every 2 days), 10 g of acerola were manually homogenized in a sterile bag with 90 mL of sterile peptone water. Serial dilutions were prepared up to 10^−4^; in sterile peptone solution for subsequent plating on 3M™ Petrifilm™ plates. Quantification of *E. coli* (CFU/g) was performed using the 3M™ Petrifilm™ E. coli/Coliform Count Plate with modified violet red bile medium and a glucuronidase activity indicator, after incubation at 42 ± 1 °C for 24 ± 2 h, where *E. coli* colonies appeared blue.

For presumptive *Salmonella* detection (presence or absence), 25 g of acerola were incubated at 41 °C for 24 h in 3M™ Salmonella Enrichment Base medium with supplement. Samples were then plated on 3M™ Petrifilm™ Salmonella Express (SALX) plates with a chromogenic medium (selective and differential for *Salmonella*). After 48 h of incubation at 41 °C, *Salmonella* colonies appeared red with yellow zones, sometimes associated with gas bubbles.

##### Weight Loss

The fruits were weighed at time 0 (immediately after coating) and every 2 days. The result was expressed as a percentage and considered the weight loss of the fruits, compared to their initial weight [25].

##### Instrumental Color

The analysis was performed as described in topic Section Color and Opacity.

##### Total Soluble Solids

The soluble solids content (SSC) was measured with a digital bench refractometer (Tecnal AR 200, New York, NY, USA) and reported in °Brix [22].

##### pH

A total of 5 g of sample was homogenized in 50 mL of distilled water, and readings were taken with a previously calibrated digital potentiometer (Tecnal TEC-7, Piracicaba, Brazil) [22].

##### Total Titratable Acidity in Molar Acid

To determine free acidity, 1 g of the sample was weighed in a 250 mL Erlenmeyer flask using an analytical balance (Shimadzu ATX 224, Kyoto, Japan). Then, 50 mL of distilled water was added and stirred until uniform. Next, 3 drops of 1% phenolphthalein were added as an indicator. Acidity was expressed as mg of molar acid per 100 g.

### 2.3. Statistical Analysis

All analyses were conducted in triplicate over two independent experiments, with results expressed as means and standard deviations. Data normality was assessed using the Shapiro–Wilk test. ANOVA, followed by Tukey’s test, was used to identify differences between means at a 5% significance level (*p* < 0.05). Data were analyzed using Sigma Stat 3.5 software (Jandel Scientific, San Jose, CA, USA) [26].

## 3. Results and Discussion

### 3.1. Characterization of Pearl Pineapple Peel Flour (PPPF)

#### 3.1.1. Physical and Technological Properties of PPPF

After processing the pineapple peels, a flour with four particle sizes was generated, as shown in Table 1. In this study, we used granulometric analysis to standardize the granule size of the flour to be used in the edible coating. The granulometry analysis, by retention rate, showed that 62.99% of the PPPF particles had a size > 16 Mesh. This result meets the desired results, as different particle sizes interfere with the solubilization process of flours and other components of the coating formulation [27]. 

Granule size has a great influence on water absorption capacity and mixing time when associated with solvents [28]. These characteristics are important for the edible coating developed in this study, since the flour needs to be completely dissolved in water so that the coating can be prepared.

The results of the PPPF technological analyses are presented in Table 2.

Regarding technological properties, PPPF presented a water solubility index (27.55%), a value higher than that found in a study [29] that analyzed flour from buriti bark (16.88%). This result indicates that PPPF has a strong bond with water, an interesting characteristic for efficient dilution of components in the preparation stage of edible coatings. 

PPPF presented a Water Absorption Index (WAI) corresponding to 2.85 g of water absorption/g of dry matter. A similar value was found in a study [30] that characterized flour from Physalis residues, and the value of 2.96 g of water absorption/g of dry matter was identified. WAI demonstrates the flour’s ability to preserve free, bound, and total water when subjected to external centrifugal gravity force or compression processes [31].

The swelling volume (SV) obtained for PPPF was equal to 10.5 mL/g, which is a higher value than that found in a study [32] with malt bagasse flour (6.11 mL/g) and lower than the value found by Pereira [33], analyzing bacuri (*Platonia insignis*) flour (23.77 mL/g). SV indicates the expansion capacity of a given element through water retention in a known amount of fiber [34]. This property influences the coating, since the greater the swelling volume, the greater the fruit’s conservation capacity, since the coating will absorb fluids [35]. 

Instrumental color analysis revealed that PPPF has a higher luminosity (L* = 76.76 ± 0.12), which is a favorable characteristic for the application of this flour in the development of edible coatings. Darker colors could restrict the use of flour in fruit coatings, affecting the acceptance of these food products.

In general, the PPPF showed a coloration tending towards yellowish red, with more positive chromaticity values a* (−1.31 ± 0.12) and b* (29.23 ± 0.00). This suggests that PPPF may be more suitable for application on fruit surfaces with this shade range, such as acerola fruits, for example. Previous studies by Silva et al. [36], who evaluated pearl pineapple peels dried in an oven with air circulation, found lower instrumental color values than those observed in this study (L* = 45.23 ± 0.25, a* = 6.06 ± 0.07 and b* = 23.69 ± 0.42). Similarly, Oliveira [37], when analyzing the PPPF of the Smooth Cayenne pineapple cultivar, obtained inferior results compared to the PPPF studied in this work (L* = 41.47 ± 3.73, a* = 3.57 ± 0.17 and b* = 22.14 ± 1.43).

#### 3.1.2. Physical and Physicochemical Characterization of PPPF

The results of the physical and physicochemical characterization of PPPF are expressed in Table 3.

The PPPF water activity (0.29) and titratable acidity (1.87%) obtained in this study differed from the values found in a previous study [38], which recorded values of 0.39 and 2.05%, respectively, for the PPPF analysis of the same variety. It is likely that these parameters directly influence the microbiological stability of the flour, given that low levels of water activity and acidic pH (high acidity) can prevent or delay the excessive multiplication of microorganisms. PPPF presented a low pH (4.1), similar to that found in another study, which recorded a pH of 4.00 [39], and higher than the value obtained in a previous study (3.8) [40], both of which evaluated the peel of the Pearl pineapple variety. Variations in pH are associated with fruit maturation, and the low value found in this study suggests that PPPF has a low potential for microbiological or enzymatic deterioration. 

The moisture content of the PPPF was 4.69%, as recommended by the Agência Nacional de Vigilância Sanitária (ANVISA) in RDC No. 263 of 2005, which establishes a maximum of 15% moisture for flours [41]. In a study where pineapple peels were oven-dried at 105 °C for 8 h, a moisture content of 9.31% was recorded [40]. Silva et al. [39] reported a moisture content of 14.10% when characterizing PPPF dried in an oven at 50 °C for 14 days, a value higher than that found in this work. ANVISA These differences can be attributed to the different methods of drying the peels in the different studies. 

The ash content (4.10%) was lower than the value found by Oliveira et al. [40] (5.51%) and higher than the result obtained by Silva et al. [39] (3.73%), both studies characterizing dehydrated pineapple peel of the same variety evaluated in this work. This content can be explained by the amount of minerals present in the PPPF, since the drying process tends to concentrate the nutrients as it reduces the water content in the raw material.

Regarding the lipid content obtained in PPPF (1.26%), a lower value (0.67%) was found in PPPF cultivar Pearl [42]. It is worth noting that several factors can influence the variations found, including the fruit ripening stage, carotenoid concentration, cultivation region, and the methodology used in the analysis. 

Oliveira [37] found protein values for the PPPF of the Smooth Cayenne variety equal to 1.45%, a lower content than that found in our study (3.07%). Oliveira et al. [40] found higher values, approximately 4.42% for the same variety studied here, thus confirming a considerable amount of protein present in the peels of this fruit. 

The carbohydrate content (78.59%) was similar to the results obtained in a previous study [37], which found 78.13% for PPPF, and higher than the 68.69% reported by Oliveira [40]. This result was expected, as fruit peels generally have higher carbohydrate contents than fresh fruit [40]. 

Higher values of total fiber (10.12%) were reported in other studies [43] and [39], which identified, respectively, 22.17% and 46.6% of crude fiber in PPPF. According to RDC No. 54 of 2012 of ANVISA, the content obtained in this study classifies PPPF as a product with high fiber content, being higher than the minimum of 6% required for foods in this category.

#### 3.1.3. Dosage of Total Phenolic Compounds

In the PPPF, the value of 320.34 mg GAE/g of total phenolic compounds was found. The result observed was lower than that reported in a study [44] that evaluated the total phenolic content of pineapple peel flour extracted using ultrasound and ethanol as a solvent, and obtained a value of 2036.80 mg GAE/g. 

Tavares and Salomão [45] evaluated the content of phenolic compounds in acerola residue and obtained values of 0.77 mg GAE/g for the in natura sample and 15.80 mg GAE/g for the lyophilized sample. The authors highlight drying by lyophilization as an efficient process, capable of concentrating the content of these substances in the sample. It is important to emphasize that the ripeness of the fruit used to prepare the flour has a direct influence on the total phenolic content evaluated. Furthermore, pineapple harvesting conditions, variety, and planting region are also capable of interfering with these results, which would justify the differences pointed out when PPPF was compared to matrices from other studies.

#### 3.1.4. Antioxidant Activity of PPPF

Overall, PPPF exhibited high antioxidant activity, with the ability to eliminate 66.11% of the ABTS^•+^ radical and 66.96% of the DPPH radical. This is likely due to the high content of total phenolic compounds in PPPF (320.34 mg EAG/g). It is important to note that several factors can influence the antioxidant activity of fruits and vegetables, including the ripening stage, species, and exposure to light, which can oxidize the compounds responsible for this activity [46].

### 3.2. Characterization of PPPF-YS Coating

#### 3.2.1. Thickness

The thickness of the PPPF-YS coating was 0.15 mm. This value may vary according to the concentration of the raw materials used in the development of the edible coating, since thicker films are composed of more viscous solutions [47]. Both the solids content and the interactions between the starch and glycerol chains can influence the increase in the film weight [48].

#### 3.2.2. Moisture Content

The moisture content found in the developed coating was 11%. It is likely that the fiber content present in the PPPF had a positive influence on this low moisture content, since fibers can reduce water retention in coatings [49].

#### 3.2.3. Colorimetric Properties and Opacity

The instrumental color parameters of the PPPF-YS edible coating on white and black backgrounds are described in Table 4. The coating presented a light hue (L* = 83.68), with a light green-yellow tendency (a* = −1.71 and b* 21.13).

The opacity of edible coatings can be influenced by the materials used in their formulation, such as biopolymers, and by additives such as plasticizers. These components affect the coating’s ability to block the passage of light, preventing undesirable reactions in the coated fruit [50]. 

The opacity found in this study for the coating was 45%. The influence of the addition of starch in coatings has gained prominence in scientific research, with an increase in opacity being identified as polysaccharide sources are added to the formulations [48,51].

Coatings with greater opacity may offer several benefits to foods, such as greater protection against light as well as reduced lipid oxidation, nutrient loss, and product deterioration [52,53,54]. In this study, we observed that the addition of yam starch possibly increased opacity, which may provide these benefits to foods coated with this formulation.

#### 3.2.4. Water Solubility

The coating showed a high water solubility index (97.21%), which can be attributed to the hydrophilic nature of glycerol. This results in higher levels of water-related properties in the coatings, facilitating the transport and interaction of water molecules with the polymer matrix [55]. 

#### 3.2.5. Mechanical Properties

Tensile strength is a crucial parameter that determines the ability of a film to withstand external forces without damage, such as cracking or tearing [23]. The PPPF-YS film presented a value of 27.77 MPa, higher than that obtained by Tafa et al. [23] (17.97 to 24.25 MPa) for films composed of various concentrations of starch agar and glycerol. Even lower tensile strength values (16.3 MPa) were reported in a study that analyzed edible film containing starch isolate from *Solanum lycocarpum* fruits (2 g), added with phenolic extract from jaboticaba (*Plinia cauliflora*) peel and glycerol [56]. These results indicate that the starch extracted from yam is more resistant to rupture. The coating also demonstrated a higher percentage of elongation at rupture (20%) than other biological films, such as the results of Tafa et al. [23], which ranged from 1.21 to 2.03% in films made of starch, agar, and glycerol. The elongation at rupture evaluates the plastic behavior and elasticity of the films.

The combination of glycerol, yam starch, and phenolic compounds present in pineapple peel flour may explain the high mobility of the film, reducing the intermolecular forces between the polymer chains and increasing the flexibility of the coating [56]. The elastic modulus obtained in this study was 1.38 Mpa. It has been reported that the addition of agar can increase the elastic modulus of coatings [23]. Furthermore, it was observed that reducing the glycerol content increases the values of this parameter, since glycerol makes the film drier and reduces intermolecular interactions between polysaccharide chains [23]. These results corroborate the findings of this research, suggesting that the addition of lower glycerol content may result in a higher elastic modulus.

#### 3.2.6. Total Phenolic Compound Content and Antioxidant Activity

The phenolic compound content found in the coating was 278.68 mg GAE/g (Table 5), a value significantly higher than that obtained in the analysis of the same formulation without the addition of pineapple peel flour, which was 85.60 mg GAE/g. This increase can be attributed to the high concentration of phenolic compounds in the pineapple peel flour, which is 320.34 mg GAE/g.

The presence of these phenolic compounds also contributes to the antioxidant activity of the coatings, acting both to inhibit and delay oxidation reactions [57]. Although the coating did not show significant values in the antioxidant activity analysis by the ABTS method, it demonstrated good antioxidant capacity by the DPPH method, with a free radical inhibition of 28.85%. The increase in antioxidant activity is directly related to the content of phenolic compounds present in the extract [56]. This is evidenced by the findings of this study, where the comparison between the coating containing PPPF-YS and the FC coating (control, without the addition of PPPF) revealed a significant antioxidant activity in the formulation with PPPF.

### 3.3. Effect of Coating on Microbiological and Physicochemical Parameters of Acerola Stored at Room and Refrigerated Temperatures

#### 3.3.1. Microbiological Evaluation

All microbiological parameters analyzed in acerola fruits coated with a formulation containing PPPF and YS (Table 6 and Table 7) complied with Normative Instruction (IN) No. 161, of 1 July 2022 [24], until the 4th day of storage at 25 ± 0.5 °C and until the 8th day of storage at 5 ± 0.5 °C. From the 6th and 10th day, when the coated acerolas were stored, respectively, at room temperature and refrigeration, some or both of the microbiological parameters recommended for quality assessment were above the maximum limit allowed by the aforementioned current legislation. These results revealed that the shelf life of acerolas, considering the microbiological quality, when coated with PPPF-YS would be up to 4 days at room temperature (25 ± 0.5 °C) and up to 8 days under refrigeration (5 ± 0.5 °C).

#### 3.3.2. Weight Loss

Citrus fruits commonly have a natural coating that is lost during the cleaning process. The coating acts as a barrier against water loss from the fruit, caused by its transpiration and respiration, which has a positive impact on weight loss [58]. Observing Figure 1, it is possible to note that weight loss occurred throughout the storage period among the three treatments, whether at room temperature or under refrigeration.

Acerolas stored at room temperature showed significant differences in weight loss between treatments (*p* < 0.05), especially on the 2nd day of storage (Figure 2A). On the 2nd day of evaluation, the lowest percentages of weight loss (*p* < 0.05) were observed for acerolas coated with either glycerol—1% (T2, 5.7%) or PPPF-YS (T3, 2.3%), when compared to the control (T1, 10%). Similarly, after the 4th day of storage at room temperature, the highest percentage of weight loss (*p* < 0.05) was observed for the control (T1, 10.7%), followed by T2 (11.5%), and when the acerola fruit was coated with PPPF-YS (T3), it presented the lowest percentage of weight loss compared to the other treatments (10%).

Regarding acerolas stored at refrigerated temperature (Figure 2B), there was a significant difference (*p* < 0.05) among all treatments. Fruits that received the PPPF-YS coating also showed lower weight loss compared to the other treatments during storage. Greater losses were recorded on the 8th day of storage (*p* < 0.05), mainly for the control treatment—T1 (10.2%), followed by treatment T2 (6%), with lower weight losses (*p* < 0.05) observed in treatment T3 (4.8%). Weight loss greater than 5% in fruits is enough to devalue the quality of the food, which leads us to conclude that storing acerolas at refrigerated temperature coated with PPPF and YS allows for a better quality of this fruit [59].

#### 3.3.3. Instrumental Color

The color of the fruit is an important sensory parameter when it comes to evaluating its shelf life and is related to the concentration of pigment, which in the case of acerola is formed by anthocyanins and carotenoids [60]. The results of the instrumental color analysis of the fruits stored at room temperature and under refrigeration are shown in Table 8 and Table 9, respectively. Regarding the fruits stored at room temperature, it was found that for the color parameter L*, which indicates the luminosity of the fruit, there was a reduction over the 4 days of storage for treatments T1 and T2 (*p* < 0.05). In contrast, when the fruit was coated with PPPF-YS, the luminosity was maintained throughout storage (*p* ≥ 0.05), indicating protection of this parameter by the coating developed with PPPF and YS, so that after 4 days it was the treatment with the highest luminosity when compared to T1 and T2 (*p* < 0.05). 

Regarding refrigerated fruits, a reduction in luminosity was observed during storage for all treatments (*p* < 0.05). After 8 days. the L value was higher for T3 (*p* < 0.05). This allows us to infer, once again, that the T3 coating was efficient in protecting the brightness of the acerola fruits when compared to the other treatments. These findings corroborate the results of Coelho [61], who reinforced that fruits coated with biodegradable ingredients present an improvement in appearance, presenting a more attractive color and brightness. 

The a* index, which is associated with pigment accumulation, did not differ significantly (*p* ≥ 0.05) between treatments when acerola fruits were stored at room temperature. The same behavior was observed for acerola fruits stored for up to 6 days at refrigeration temperature (*p* ≥ 0.05). However, a reduction in this parameter was observed for all treatments during storage, both at room temperature and under refrigeration (*p* < 0.05). The PPPF-YS coating protected this color parameter more during refrigerated storage, considering that T3 presented higher a* values compared to T1 and T2 (*p* < 0.05), after 8 days of storage.

A reduction in color b* (*p* < 0.05) was also observed for acerola fruits stored both at room temperature and under refrigeration. The reductions were smaller (*p* < 0.05) when acerola fruits were coated with PPPF-YS (T3) and stored under refrigeration, demonstrating once again the potential of this coating to protect the color parameters of acerola fruits under this condition. 

At the end of storage, both at room temperature and at refrigerated temperature, the acerola fruits showed a predominance of red over yellow color, especially the fruits coated with T3. This characteristic is important from a sensory point of view, in which acerolas are expected to be red until the end of storage. It is possible that the higher values of total phenolic compounds and antioxidant activity detected in the T3 coating contributed to protection against oxidation processes of the pigments present in the acerola fruits, contributing to the preservation of their color throughout storage. 

Surface dehydration in fruits may also be linked to color changes throughout the post-harvest period, which can be seen in this study, since the acerolas in the negative control group (without coating) in which greater color variation was observed in the values of L, a* and b*, also showed greater weight loss [62].

#### 3.3.4. Analysis of Total Soluble Solids, pH, and Total Titratable Acidity

The changes in the physical–chemical parameters of acerola fruits subjected to different coating treatments (T1, T2, and T3), stored at room temperature and refrigerated, are described in Table 10 and Table 11, respectively. The soluble solids content is a considerable indicator of fruit ripening, since it is associated with the sweetness content that increases during storage, because of the accumulation of sugars [63]. In this study, the total soluble solids content (TSS) remained throughout storage at room temperature for T2 and T3 (*p* ≥ 0.05) and reduced in T1 (*p* < 0.05) from the 2nd day of storage. 

When the acerolas subjected to the same coating treatments were stored at refrigerated temperature, maintenance of the SST content was observed in T2 (*p* ≥ 0.05), and a reduction was observed in T1 and T3 (*p* < 0.05). In this study, there was no influence of the type of coating on the SST content on any of the days of refrigerated storage (*p* ≥ 0.05). 

For the pH variable, no significant difference (*p* ≥ 0.05) was observed either between treatments or between times when the acerolas were stored at room temperature (Table 10). All treatments were in accordance with current legislation, which highlights a minimum pH value of 2.80 for acerolas [64]. Results close to those found in this study (between 3.15 and 3.40) were obtained by Rocha et al. [65], although they observed a reduction in pH values at the end of the storage period at room temperature of guavas coated with cassava and cornstarch. 

Regarding the fruits stored at refrigeration temperature (Table 11), it was possible to note a significant increase (*p* < 0.05) in pH after 8 days of storage in group T2 (from 3.42 ± 0.04 to 3.61 ± 0.02) and maintenance of pH in T1 and T3 (*p* ≥ 0.05), compared to the first day of storage. It is important to highlight that low pH values in acerolas are interesting, as they indicate better post-harvest conservation, allowing longer storage as they limit the development of microorganisms [66]. In our study, the evaluations were carried out up to the 8th day of refrigerated storage, considering that after 10 days the fruits presented microbiological parameters above those permitted by current legislation for the identity and quality standard of acerolas [24]. 

Regarding the titratable acidity content (TAC), measured in % citric acid, the values of refrigerated and non-refrigerated fruits ranged from 1.27 to 2.04% during storage at room temperature and refrigeration. With the exception of the control group (T1), the acidity values were maintained during storage at room temperature (*p* ≥ 0.05). Furthermore, there was no statistical difference between the different coating treatments (*p* ≥ 0.05) when the acerolas were stored at 25 ± 0.5 °C. Differently from what was observed in this study, Rodrigues et al. [27] evaluated acerola fruits coated with sunflower oil, cassava starch, and colorless gelatin and stored at room temperature and observed increased acidity values after 7 days of storage. In our research, we saw that during refrigerated storage there was no change in the TAC of acerola fruits stored at refrigerated temperature (*p* ≥ 0.05).

Coatings can delay changes in pH, titratable acidity, and total soluble solids (TSS) in fruits, slowing down the ripening process and, consequently, senescence. This is associated with the semipermeability that the coating confers to the fruit, modifying the internal atmosphere by altering the concentrations of O_2_ and CO_2_. In the study in question, it was seen that when the acerolas were coated (T2 or T3), the changes during storage, whether at room or refrigerated temperature, tended to be smaller or were delayed, with no significant changes throughout the shelf life for most of the quality parameters studied, demonstrating the potential of coatings for the post-harvest control of acerolas [67].

## 4. Conclusions

In this study, we developed a novel coating using pearl pineapple peel flour and yam starch (PPPF-YS) for preserving acerola fruits during storage at both room and refrigerated temperatures. The PPPF-YS coating exhibited desirable attributes such as a light color, low moisture content, high water solubility, and good mechanical properties—crucial for maintaining the quality of coated acerola fruits. Notably, the addition of pearl pineapple peel flour enhanced the phenolic content and antioxidant activity compared to the flour-free formulation. The coated acerola fruits met microbiological standards for up to 4 days at room temperature and up to 8 days under refrigeration. The PPPF-YS coating effectively reduced weight loss preserved the fruits’ light and shiny appearance, and delayed ripening, particularly under refrigeration. These results highlight the potential of using sustainable technologies to reduce costs and environmental impact using agri-food waste in edible coatings. This research supports Sustainable Development Goal (SDG) 12 for sustainable consumption and production, and SDG 2 for ending hunger, achieving food security, and promoting sustainable agriculture.

## Figures and Tables

**Figure 1 foods-13-02873-f001:**
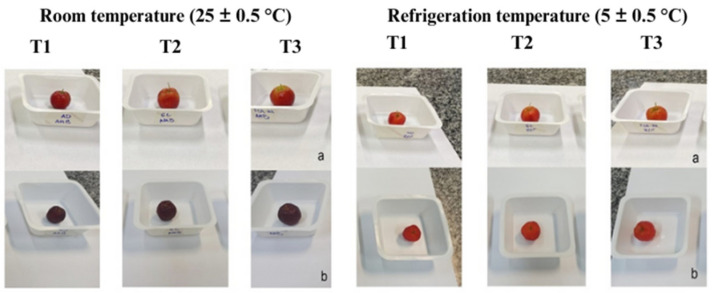
Effect of coatings T1—negative control (fruits without coating); T2—positive control (fruits coated with 1% glycerol) and T3—fruits coated with pearl pineapple peel flour-yam starch (PPPF-YS) on weight loss and instrumental color of acerola fruits at times 0 (**a**) and 4 (**b**) days of storage at room temperature (25 ± 0.5 °C) and at times 0 (**a**) and 8 (**b**) days of storage at refrigeration temperature (5 ± 0.5 °C).

**Figure 2 foods-13-02873-f002:**
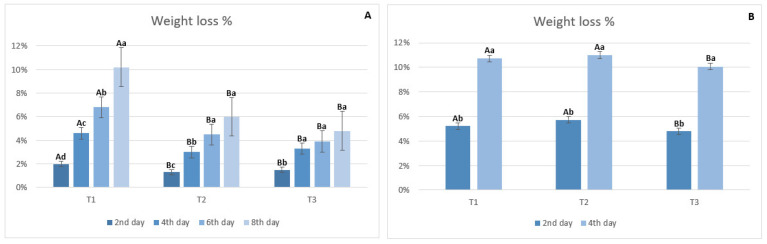
Weight loss of acerolas stored at room temperature (25 ± 0.5 °C) (**A**) and under refrigeration (5 ± 0.5 °C) (**B**) with different coatings: T1—negative control (fruits without coating); T2—positive control (fruits coated with 1% glycerol) and T3—fruits coated with pearl pineapple peel flour-yam starch (PPPF-YS). a–d Different lowercase letters within the same treatment indicate significant differences between storage times according to the Student’s *t*-test or Tukey’s test (*p* < 0.05). A–B Different uppercase letters within the same storage time indicate significant differences between treatments according to Tukey’s test (*p* < 0.05).

**Table 1 foods-13-02873-t001:** Granulometric analysis of pearl pineapple peel flour (PPPF).

Granulometry				
Mesh	Size (mm)	Initial Weight (g)	Final Weight (g) %	Retention
16	1.19	456.62 ± 0.32	519.61 * ± 0.40	62.99 ± 0.08
18	1.00	458.35 ± 0.10	464.09 * ± 0.19	5.74 ± 0.09
20	0.84	513.07 ± 0.14	524.34 * ± 0.24	11.32 ± 0.10
35	0.50	379.33 ± 0.23	385.85 * ± 0.65	6.52 ± 0.43
Base	<0.07	387.07 ± 0.17	402.58 * ± 0.28	15.51 ± 0.11

* Values in relation to the initial weight of the sample (100 g).

**Table 2 foods-13-02873-t002:** Water Absorption Index (WAI) and swelling volume (SV) (Mean ± SD) of pearl pineapple peel flour (PPPF).

Parameters	PPPF
WAI (g absorbed/g dry weight)	2.85 ± 0.14
Water solubility (%)	27.55 ± 2.95
SV (mL/g dry weight)	10.5 ± 0.00

**Table 3 foods-13-02873-t003:** Physical and physicochemical characterization (Mean ± SD) of pearl pineapple peel flour (PPPF).

Parameters	PPPF
Water activity	0.29 ± 0.00
Titratable acidity (%)	1.87 ± 0.04
pH	4.18 ± 0.16
Moisture (%)	4.69 ± 0.31
Ash (%)	4.10 ± 0.00
Lipids (%)	1.26 ± 0.23
Proteins (%)	3.07 ± 0.00
Carbohydrates (%)	78.59 ± 0.03
Total fiber (%)	10.12 ± 0.00

**Table 4 foods-13-02873-t004:** Instrumental color presented by pearl pineapple peel flour-yam starch (PPPF-YS) coating.

Parameters	White Background	Black Background
Luminosity (L*)	83.68 ± 0.78	37.92 ± 0.36
a*	−1.71 ± 0.11	−1.06 ± 0.16
b*	21.13 ± 2.39	6.88 ± 0.43

**Table 5 foods-13-02873-t005:** Antioxidant activity and dosage of phenolic compounds (PC) of coatings: pearl pineapple peel flour-yam starch (PPPF-YS), and coating without the addition of pineapple peel flour (FC).

Coating	DPPH (%)	ABTS (%)	PC (mg GAE/g)
PPPF-YS	28.85 ± 0.27	<LOD	278.68 ± 0.45
FC	<LOD	<LOD	85.60 ± 0.56

LOD: below the detection limit.

**Table 6 foods-13-02873-t006:** Microbiological evaluation of acerolas coated with pearl pineapple peel flour-yam starch (PPPF-YS) during storage under room temperature (25 ± 0.5 °C).

Microbial Parameter	Treatments	Time (Days)			Microbiological Standards *
		0	2	4	
*E. coli*	T1	Abs	Abs	Abs	5 × 10^3^ UFC/g
	T2	Abs	Abs	Abs	
	T3	Abs	Abs	Abs	
*Salmonella* spp.	T1	Abs	Abs	Abs	Abs
	T2	Abs	Abs	Abs	
	T3	Abs	Abs	Abs	

Treatments: T1—negative control (fruits without coating); T2—positive control (fruits coated with 1% glycerol); and T3—fruits coated with PPPF-YS. * Normative Instruction (IN) No. 161, of 1 July 2022 [24]. Abs: absent.

**Table 7 foods-13-02873-t007:** Microbiological evaluation of acerolas coated with pearl pineapple peel flour-yam starch (PPPF-YS) during storage under refrigeration temperature (5 ± 0.5 °C).

Microbial Parameter	Treatments	Time (Days)					Microbiological Standards *
		0	2	4	6	8	
*E. coli*	T1	Abs	Abs	Abs	Abs	Abs	5 × 10^3^ UFC/g
	T2	Abs	Abs	Abs	Abs	Abs	
	T3	Abs	Abs	Abs	Abs	Abs	
*Salmonella* spp.	T1	Abs	Abs	Abs	Abs	Abs	Abs
	T2	Abs	Abs	Abs	Abs	Abs	
	T3	Abs	Abs	Abs	Abs	Abs	

Treatments: T1—negative control (fruits without coating); T2—positive control (fruits coated with 1% glycerol); and T3—fruits coated with PPPF-YS. * Normative Instruction (IN) No. 161, of 1 July 2022 [24]. Abs: absent.

**Table 8 foods-13-02873-t008:** Effect of coating with pearl pineapple peel flour-yam starch (PPPF-YS) on the instrumental color of acerolas stored at room temperature (25 ± 0.5 °C).

Treatments	Storage Days (25 ± 0.5 °C).		
	0	2	4
L*			
T1	26.63 ± 4.72 ^Aa^	24.22 ± 2.29 ^Aa^	18.21 ± 1.33 ^Bb^
T2	31.87 ± 2.08 ^Aa^	23.15 ± 4.04 ^Ba^	17.53 ± 1.98 ^Bb^
T3	30.59 ± 3.60 ^Aa^	27.48 ± 0.75 ^Aa^	27.76 ± 2.96 ^Aa^
a*			
T1	34.60 ± 2.81 ^Aa^	30.05 ± 3.02 ^Aa^	21.36 ± 1.72 ^Ba^
T2	36.79 ± 0.97 ^Aa^	27.64 ± 3.82 ^Ba^	21.72 ± 0.76 ^Ca^
T3	36.91 ± 1.47 ^Aa^	25.93 ± 1.55 ^Ba^	19.39 ± 1.62 ^Ca^
b*			
T1	11.95 ± 1.56 ^Ab^	8.87 ± 0.72 ^Ba^	5.06 ± 0.16 ^Ca^
T2	14.79 ± 1.33 ^Aab^	8.46 ± 1.43 ^Ba^	5.86 ± 0.65 ^Ba^
T3	15.48 ± 0.86 ^Aa^	8.03 ± 0.49 ^Ba^	5.57 ± 0.34 ^Ca^

Treatments: T1—negative control (fruits without coating); T2—positive control (fruits coated with 1% glycerol); and T3—fruits coated with PPPF-YS. a–b Means and standard deviations followed by different lowercase letters in the same column differ from each other by Tukey’s test (*p* < 0.05), among treatments. A–C Means and standard deviations followed by different uppercase letters in the same row differ from each other by Tukey’s test (*p* < 0.05), among storage times.

**Table 9 foods-13-02873-t009:** Effect of coating with pearl pineapple peel flour-yam starch (PPPF-YS) on the instrumental color of acerolas stored at refrigeration temperature (5 ± 0.5 °C).

Treatments	Storage Days (5 ± 0.5 °C)				
	0	2	4	6	8
L*					
T1	31.04 ± 3.68 ^Aa^	27.04 ± 3.49 ^ABa^	21.56 ± 5.45 ^ABa^	25.25 ± 2.91 ^ABa^	17.79 ± 0.04 ^Bb^
T2	30.26 ± 6.22 ^Aa^	27.19 ± 3.88 ^ABa^	19.70 ± 2.48 ^ABa^	22.15 ± 2.12 ^ABa^	19.23 ± 0.08 ^Bb^
T3	32.03 ± 1.80 ^Aa^	33.00 ± 3.48 ^Aa^	25.89 ± 3.27 ^ABa^	25.97 ± 3.04 ^ABa^	22.10 ± 0.08 ^Ba^
a*					
T1	34.71 ± 2.08 ^Aa^	30.56 ± 2.99 ^ABa^	23.43 ± 5.87 ^Ba^	26.81 ± 3.26 ^ABa^	18.25 ± 0.04 ^Bb^
T2	34.34 ± 3.61 ^Aa^	30.33 ± 1.52 ^ABa^	25.18 ± 0.67 ^Ba^	27.40 ± 2.68 ^Ba^	18.53 ± 0.04 ^Cb^
T3	29.27 ± 4.62 ^Aa^	31.23 ± 2.53 ^Aa^	30.23 ± 1.05 ^Aa^	28.45 ± 2.06 ^ABa^	21.21 ± 0.01 ^Ba^
b*					
T1	13.90 ± 1.49 ^Aa^	12.78 ± 0.32 ^ABa^	9.96 ± 3.74 ^ABa^	11.69 ± 1.69 ^ABa^	7.72 ± 0.05 ^Bb^
T2	13.40 ± 2.64 ^Aa^	11.95 ± 2.03 ^ABa^	10.20 ± 1.09 ^ABa^	11.55 ± 1.53 ^ABa^	7.73 ± 0.04 ^Bb^
T3	14.91 ± 0.56 ^Aa^	1511 ± 1.02 ^Aa^	13.57 ± 0.91 ^ABa^	12.13 ± 0.90 ^BCa^	10.34 ± 0.03 ^Ca^

Treatments: T1—negative control (fruits without coating); T2—positive control (fruits coated with 1% glycerol); and T3—fruits coated with PPPF-YS. a–b Means and standard deviations followed by different lowercase letters in the same column differ from each other by Tukey’s test (*p* < 0.05), among treatments. A–C Means and standard deviations followed by different uppercase letters in the same row differ from each other by Tukey’s test (*p* < 0.05), among storage times.

**Table 10 foods-13-02873-t010:** Physicochemical characteristics of acerolas coated with pearl pineapple peel flour-yam starch (PPPF-YS) stored at room temperature (25 ± 0.5 °C).

Treatments	Storage Days (25 ± 0.5 °C)		
	0	2	4
Total soluble solids (°Brix)			
T1	11.95 ± 0.92 ^Aa^	9.65 ± 0.21 ^Ba^	9.20 ± 0.00 ^Ba^
T2	9.45 ± 1.06 ^Aa^	9.65 ± 0.21 ^Aa^	8.00 ± 0.28 ^Ab^
T3	10.15 ± 0.50 ^Aa^	9.35 ± 0.21 ^Aa^	8.85 ± 0.07 ^Aa^
pH			
T1	3.44 ± 0.01 ^ABa^	3.35 ± 0.04 ^Ba^	3.56 ± 0.06 ^Aa^
T2	3.43 ± 0.04 ^Aa^	3.37 ± 0.02 ^Aa^	3.47 ± 0.11 ^Aa^
T3	3.44 ± 0.01 ^Aa^	3.27 ± 0.02 ^Aa^	3.42 ± 0.16 ^Aa^
Total titratable acidity (% citric acid)			
T1	1.48 ± 0.04 ^Ba^	1.62 ± 0.13 ^Ba^	2.01 ± 0.04 ^Aa^
T2	1.40 ± 0.28 ^Aa^	1.55 ± 0.23 ^Aa^	1.68 ± 0.21 ^Aa^
T3	1.37 ± 0.06 ^Aa^	1.40 ± 0.18 ^Aa^	1.76 ± 0.04 ^Aa^

Treatments: T1—negative control (fruits without coating); T2—positive control (fruits coated with 1% glycerol); and T3—fruits coated with PPPF-YS. a–b Means and standard deviations followed by different lowercase letters in the same column differ from each other by Tukey’s test (*p* < 0.05), among treatments. A–B Means and standard deviations followed by different uppercase letters in the same row differ from each other by Tukey’s test (*p* < 0.05), among storage times.

**Table 11 foods-13-02873-t011:** Physicochemical characteristics of acerolas coated with pearl pineapple peel flour-yam starch (PPPF-YS) stored at refrigeration temperature (5 ± 0.5 °C).

Treatments	Storage Days (5 ± 0.5 °C)				
	0	2	4	6	8
Total soluble solids (°Brix)					
T1	11.94 ± 0.92 ^Aa^	9.35 ± 0.35 ^Ba^	8.10 ± 0.28 ^Ba^	9.00 ± 0.14 ^Ba^	9.95 ± 0.07 ^Ba^
T2	9.44 ± 1.06 ^Aa^	9.55 ± 0.35 ^Aa^	9.50 ± 0.56 ^Aa^	9.15 ± 0.35 ^Aa^	9.70 ± 0.28 ^Aa^
T3	10.14 ± 0.49 ^Aa^	9.05 ± 0.07 ^ABa^	9.05 ± 0.50 ^ABa^	8.55 ± 0.21 ^Ba^	9.30 ± 0.14 ^ABa^
pH					
T1	3.43 ± 0.01 ^Aa^	3.51 ± 0.01 ^Aa^	3.38 ± 0.11 ^Aa^	3.49 ± 0.04 ^Aa^	3.41 ± 0.08 ^Aa^
T2	3.42 ± 0.04 ^Ba^	3.43 ± 0.01 ^Ba^	3.50 ± 0.07 ^ABa^	3.43 ± 0.01 ^Ba^	3.61 ± 0.02 ^Aa^
T3	3.43 ± 0.01 ^Aa^	3.49 ± 0.07 ^Aa^	3.47 ± 0.05 ^Aa^	3.52 ± 0.06 ^Aa^	3.56 ± 0.09 ^Aa^
Total titratable acidity (% citric acid)					
T1	1.47 ± 0.04 ^Aa^	1.82 ± 0.02 ^Aa^	1.56 ± 0.12 ^Aa^	1.65 ± 0.00 ^Aa^	1.56 ± 0.21 ^Aa^
T2	1.39 ± 0.27 ^Aa^	1.27 ± 0.09 ^Ab^	1.72 ± 0.09 ^Aa^	1.74 ± 0.04 ^Aa^	1.53 ± 0.09 ^Aa^
T3	1.36 ± 0.05 ^CDa^	2.04 ± 0.10 ^Aa^	1.78 ± 0.08 ^ABa^	1.68 ± 0.04 ^BCa^	1.33 ± 0.09 ^Da^

Treatments: T1—negative control (fruits without coating); T2—positive control (fruits coated with 1% glycerol); and T3—fruits coated with PPPF-YS. a–b Means and standard deviations followed by different lowercase letters in the same column differ from each other by Tukey’s test (*p* < 0.05), among treatments. A–D Means and standard deviations followed by different uppercase letters in the same row differ from each other by Tukey’s test (*p* < 0.05), among storage times.

## Data Availability

The original contributions presented in the study are included in the article, further inquiries can be directed to the corresponding author.

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
