# Peer review of "Development of Edible Coatings Based on Pineapple Peel (Ananas Comosus L.) and Yam Starch (Dioscorea alata) for Application in Acerola (Malpighia emarginata DC)"

_foods, 2024, doi:10.3390/foods13182873_

Round 1
Reviewer 1 Report
Comments and Suggestions for Authors
Review on manuscript: foods-3201928
Development of edible coatings based on pineapple peel (Ananas Comosus L.) and yam starch (Dioscorea alata) for application in acerola (Malpighia emarginata DC))
by Maria Brígida Fonseca Galvão, Thayza Christina Montenegro Stamford, Flávia Alexsandra Belarmino Rolim de Melo, Gerlane Souza de Lima, Carlos Eduardo Vasconcelos de Oliveira, Ingrid Luana Nicácio de Oliveira, Rita de Cassia de Araujo Bidô, Maria Manuela Estevez Pintado, Maria Elieidy Gomes de Oliveira, & Tania Lucia Montenegro Stamford*
Research paper
This manuscript deceloped an edible coatings based on pineapple peel and yam starch, and investigated its application in acerola. Overall, it was of great significance and written well. However, some modifications are still required for futher quality improvement of the whole manuscript.
Detailed recommendation:
-Title: "(Malpighia emarginata DC))" should be "(Malpighia emarginata DC)".
-Introduction: two many short paragraphs, which are better to be incorporated into 3-4 long paragraphs.
-Materials and methods: it's very detailed but two long. I think this part can be properly focused.
-Tables 1 & 4: the standard deviations are missing.
-Figure 1: it can be removed.
-Figure 3: the significance of data difference should be marked.
-Conclusions: this part is usually one brief and focused paragraph.
-References: two many cited publications. Normally 30-50 references are appropriate for a research paper.
-The English-wtiring is good enough and I think only minor language editing is required.
Comments on the Quality of English LanguageThe English-wtiring is good enough and I think only minor language editing is required.
Author Response
Point 1: Título: "(Malpighia emarginata DC))" deve ser "(Malpighia emarginata DC)".
Response 1: We appreciate your suggestion. We made the change; please refer to line 4.
Point 2: Introduction: two many short paragraphs, which are better to be incorporated into 3-4 long paragraphs.
Response 2: We appreciate your suggestion. To address the suggestion and improve the structure of the introduction, we combined the shorter paragraphs into larger sections and created a more cohesive narrative. See changes made on page 1, lines 50 to 90.
Point 3: Materials and methods: it's very detailed but two long. I think this part can be properly focused.
Response 3: Thank you for your feedback. We appreciate your suggestion to streamline the "Materials and Methods" section. We have revised this section to focus on the essential details and improve clarity. The updated version is more concise while retaining all critical information.
Point 4: Tables 1 & 4: the standard deviations are missing.
Response 4: We appreciate your observation. We have included the standard deviations. Please refer to Tables 1 and 4.
Point 5: Figure 1: it can be removed.
Response 5: Thank you for your suggestion. We have removed Figure 1.
Point 6: Figure 3: the significance of data difference should be marked.
Response 6: Thank you for your suggestion. We included the significance of the data in the Figure. Please refer Figure 2.
Point 7: Conclusions: this part is usually one brief and focused paragraph.
Response 7: Thank you for your feedback. We have revised the conclusion to be more concise and focused, consolidating the key findings into a single brief paragraph. The updated conclusion reflects the essential points of the study while adhering to the standard format for this section.
Point 8: References: two many cited publications. Normally 30-50 references are appropriate for a research paper.
Response 8: We appreciate your observation. However, the number of references cited is due to the suggestion of another reviewer who requested additional citations. We aimed to include a reasonable number of references to address the previous reviewer’s request. We have reviewed other manuscripts and found that the number of references we included is consistent with those in articles published in the journal Foods.
Point 9: The English-wtiring is good enough and I think only minor language editing is required.
Response 9: Thank you for your feedback. We perform small edits to ensure clarity and improve the overall quality of the manuscript.

Reviewer 2 Report
Comments and Suggestions for Authors
The topic of the article entitled "Development of edible coatings based on pineapple peel (Ananas Comosus L.) and yam starch (Dioscorea alata) for application in acerola (Malpighia emarginata DC)" is interesting and relevant for vegetable processors, because increasing the shelf life and availability of acerola can have health benefits for consumers and financial benefits for fruit producers. However, some changes are necessary:
- ''The total phenolic content of 39 the coating was 278.68 ± 0.45'', is missing the units for polyphenols, lines 39-40
- "The analysis of the water solubility of the flour was performed." indent of paragraph is different, line 159
- "gallic acid equivalent per gram, 190 of sample. (mg EAG/g)." delete the dot after sample; line 190-191
Also, what about GAE instead of EAG, all the text.
- "The ABTS (2,2’-azino-bis(3-ethylbenzothiazolin)6-sulfonic acid) radical", indent of paragraph is different, line 196
- ABTS●+, is correct? line 194
- ABTS●+ radical, is correct? line 198
- For the DPPH radical scavenging assay●, the dot is necessary? line 212
- P.A. methanol, what does it mean? line 213
- DPPH● radical, is correct? line 217
- humidity determination takes 24 h or until constant mass? line 233-236
- "ABTS●+ radical and 66.96% of the DPPH● radical" is correct? line 451
Author Response
Point 1: The total phenolic content of 39 the coating was 278.68 ± 0.45'', is missing the units for polyphenols, lines 39-40.
Response 1: Thank you for your observation. We have included the unit. Please refer to line 40.
Point 2: The analysis of the water solubility of the flour was performed." indent of paragraph is different, line 159.
Response 2: We appreciate your observation. We have adjusted the paragraph indentation. Please refer to line 222.
Point 3: gallic acid equivalent per gram, 190 of sample. (mg EAG/g)." delete the dot after sample; line 190-191.
Response 3: We appreciate your observation. We have removed the dot after "sample". Please refer to line 265.
Point 4: Also, what about GAE instead of EAG, all the text.
Response 4: Thank you for your suggestion. We have replaced it with GAE throughout the text.
Point 5: The ABTS (2,2’-azino-bis(3-ethylbenzothiazolin)6-sulfonic acid) radical", indent of paragraph is different, line 196.
Response 5: We appreciate your observation. We have adjusted the paragraph indentation. Please refer to line 275.
Point 6: ABTS●+, is correct? line 194.
Response 6: The correct notation is ABTS•+ (with a single dot for the radical and a superscript plus sign). The dot represents the radical nature of the ion, and the superscript plus sign indicates its positive charge. We have made the adjustments throughout the text. Other manuscripts published in Foods use the same notation. Please refer to: https://doi.org/10.3390/foods11233814; https://doi.org/10.3390/foods11233876; https://doi.org/10.3390/foods11233889; https://doi.org/10.3390/foods11233870.
Point 7: ABTS●+ radical, is correct? line 198.
Response 7: The correct notation is ABTS•+ (with a single dot for the radical and a superscript plus sign). The dot represents the radical nature of the ion, and the superscript plus sign indicates its positive charge. We have made the adjustments throughout the text. Other manuscripts published in Foods use the same notation. Please refer to: https://doi.org/10.3390/foods11233814; https://doi.org/10.3390/foods11233876; https://doi.org/10.3390/foods11233889; https://doi.org/10.3390/foods11233870.
Point 8: For the DPPH radical scavenging assay●, the dot is necessary? line 212.
Response 8: We apologize. We have removed the ● after "assay." Please see line 300.
Point 9: P.A. methanol, what does it mean? line 213.
Response 9: We apologize. We actually meant to refer to A.R. (Analytical Reagent) grade. We have made the correction throughout the text. Please see lines 278 and 301.
Point 10: DPPH● radical, is correct? line 217.
Response 10: Thank you for your observation. We have removed the "●" symbol.
Point 11: humidity determination takes 24 h or until constant mass? line 233-236.
Response 11: In this study, we subjected the edible coating to oven drying at a controlled temperature of 105 °C for moisture determination. Moisture content determination can be performed over 24 hours, especially with an oven drying method at a controlled temperature such as 105 °C. A 24-hour period is common in drying procedures to ensure that the sample reaches a constant mass and that all moisture has been completely removed. While the exact duration may vary depending on the material and specific method, 24 hours is a standard practice to ensure accurate and reproducible results. To improve clarity, we have added this information. Please see lines 330 to 331.
Point 12: ABTS●+ radical and 66.96% of the DPPH● radical" is correct? line 451.
Response 12: Thank you for your feedback. We have revised the text for clarity and adjusted the symbols for the ABTS and DPPH radicals. Please refer to lines 604 to 609.
